# Therapeutic and Nutraceutical Potential of *Sargassum* Species: A Narrative Review

**DOI:** 10.3390/md23090343

**Published:** 2025-08-28

**Authors:** Alejandra Torres-Narváez, Andrea Margarita Olvera-Ramírez, Karen Castaño-Sánchez, Jorge Luis Chávez-Servín, Tércia Cesária Reis de Souza, Neil Ross McEwan, Roberto Augusto Ferriz-Martínez

**Affiliations:** 1Laboratorio de Biomédica y en Alimentos Funcionales, Facultad de Ciencias Naturales, Universidad Autónoma de Querétaro, Avenida de las Ciencias S/N, Juriquilla, Querétaro C.P. 76230, Mexico; alejandra.torres@uaq.edu.mx (A.T.-N.); kcastano14@alumnos.uaq.mx (K.C.-S.); jorge.chavez@uaq.mx (J.L.C.-S.); 2Cuerpo Académico Nutrición, Salud y Producción Animal, Facultad de Ciencias Naturales, Universidad Autónoma de Querétaro, Avenida de las Ciencias S/N, Juriquilla, Querétaro C.P. 76230, Mexico; andrea.olvera@uaq.mx (A.M.O.-R.); tercia@uaq.mx (T.C.R.d.S.); 3Cuerpo Académico Investigación Biomédica y en Alimentos Funcionales, Facultad de Ciencias Naturales, Universidad Autónoma de Querétaro, Avenida de las Ciencias S/N, Juriquilla, Querétaro C.P. 76230, Mexico; 4SRUC School of Veterinary Medicine, Scotland’s Rural College, Craibstone Estate, Aberdeen AB21 9YA, Scotland, UK; neil.mcewan@sruc.ac.uk

**Keywords:** macroalgae, therapeutic potential, nutraceutical potential, *Sargassum*

## Abstract

In the face of agricultural and environmental crises, the ocean and its diverse abundance of species have garnered attention as sources of beneficial compounds for humans, offering sustainable solutions across various sectors with minimal environmental impact. *Sargassum*, a genus of macroalgae, has long been used in alternative medicine and culinary applications. This genus encompasses a wide variety of species, many of which contain bioactive compounds with significant therapeutic potential that remain under investigation. Some *Sargassum* species not only represent a valuable resource but also pose challenges due to their overgrowth, making their utilization both essential and strategic. In this narrative review we highlight many of the major physiological effects of these compounds, concentrating on their promising role in addressing global challenges.

## 1. Introduction

Seaweed comprises algae which have been used in different industries and cultures as foods, cosmetics and nutraceuticals. Their uses have been described in a number of different parts of the world including Asia (e.g., China, Japan and Korea), Europe (e.g., Austria, Germany and Britain) and North America (e.g., United States and Canada), among others [1,2]. Seaweed can be produced or collected in its natural habitat [3] and among the seaweeds which are consumed are the macroalgae nori, kombu and wakame [1]. Due to the pressures placed on land for agricultural use, alternatives are regularly being sought which do not require land for growth. In this context, seaweed is an important candidate for meeting this requirement as it does not require land for growth. Moreover, it does not place pressure on areas of freshwater as it grows in the sea; it can obtain the necessary nutrients from the sea and grow rapidly [4,5].

In addition to its environmental benefits, there are also health benefits [6]. For example, across the various different species of macroalgae, nearly 3000 natural compounds have been identified [7]. Specifically in the context of seaweed, these organisms have been identified as sources of functional metabolites with a range of potential roles. These include activities which are anti-inflammatory, antiviral and antitumor, as well as prebiotic roles, having effects on chronic non-communicable diseases, as well as neuroprotective and anticancer properties [2,8,9].

This paper provides a narrative review of the potential therapeutic and nutraceutical importance of seaweed and algae, with a particular emphasis on extracts isolated from species which are members of the genus *Sargassum* (Phaeophyceae). Regarding these therapeutic and nutraceutical applications, eight aspects are addressed: (1) anti-inflammatory activity; (2) antioxidant activity; (3) antiviral activity; (4) antibacterial activity; (5) prebiotic activity; (6) chronic non-communicable diseases; (7) anticancer activity and (8) neuroprotective activity.

## 2. Algae

Algae are photosynthetic autotrophic organisms that first appeared on Earth around 1.6–1.7 billion years ago and are currently responsible for around half of the fixation of carbon dioxide which takes place on the planet [10,11]. They can be broadly classified into two groups according to their size: macroalgae (some of which can reach sizes of up to 60 m in length) or microalgae (which range from as little as 1 mm to several cm in length) [12]. In addition to classification based on size, they can also be classified by the type of pigments they have. There are three types of macroalgae: green algae (Chlorophytes), red algae (Rhodophytes) and brown algae (Heterokontophyta and Phaeophyceae). In green algae, pigments such as chlorophyll, β carotenes and xanthophylls predominate. Red algae are dominated by xanthophylls, phycoerythrin and phycocyanin. Brown algae are abundant in β-carotenes, xanthophylls and fucoxanthins [13].

Globally, there are around 9200 reported species of macroalgae, comprising approximately 6000 red algae, 2000 brown algae and 1200 green algae [14]. Their composition is influenced by different factors such as habitat, latitude, seasons of the year, stage of maturity, temperature, conditions at the time of selecting the sample, availability of macronutrients (nitrogen, potassium and phosphorus), pH, gases (primarily O_2_ and CO_2_), light intensity and their associated microbiota [15,16,17]. It should be noted that, even between species, the morphology can vary depending on the conditions and different seasons [18]. Generally, macroalgae are composed of carbohydrates, lipids, proteins, minerals, enzymes, antioxidants, secondary metabolites such as phytochemicals and vitamins A, C, E, B_1_, B_2_, B_3_, B_5_, B_6_, B_7_ and B_9_ [14,17]. Likewise, thanks to symbiosis with microorganisms, these are rich sources of vitamin B_12_ [10]. In general, green and red algae have a higher content of B vitamins, pantothenic acid, folic acid and short chain fatty acids [14]. As for minerals, these represent between 20 and 50% of the dry weight. The minerals present include potassium, calcium, sodium, phosphorus, copper, iron, selenium, manganese, zinc, magnesium, chromium and iodine, the last of which is particularly high in brown algae [3].

Like terrestrial plants, algae do not have an immune system and rely on the production of phytochemicals to protect themselves from factors such as predators, tissue damage, ultraviolet light, desiccation and salinity [16]. Macroalgae produce a wide variety of metabolites of different types relative to those present in terrestrial plants. These include metabolites such as sulfated polysaccharides and fucoxanthin [14]. Likewise, diterpenes, sterols, carotenoids, phenolic compounds and alkaloids are present [19]. It has been shown that brown algae contain concentrations of polyphenols up to 10 times higher than red and green algae [20].

Specific to seaweeds, their main components are carbohydrates, which constitute around 50–60% of their weight [21]. These carbohydrates are classified as storage, structural and functional forms, with most of the structural functions relating to the cell wall [22]. The structural polysaccharides consist of insoluble compounds (e.g., cellulose, xylans and mannans), soluble sulfated polysaccharides (e.g., carrageenan, fucoidan and ulvan) and non-sulfated polysaccharides (e.g., laminaran and agarose) [11]. Fucoidans are mainly synthesized by brown algae, carrageenan by red algae and ulvans by green algae [23]. Green and red macroalgae are generally considered to be rich in carbohydrates, while brown macroalgae are higher in soluble fiber. In humans, having a source of dietary soluble fiber contributes to a reduction in blood glucose and plasma cholesterol concentrations [15]. In terms of algal consumption, in some Asian countries this can be up to 14 g of algae per day, making a considerable contribution to the daily requirement of 20–25 g of fiber [1].

Algae have a low lipid content, typically in the range of 1.5–3%, with most of the lipids containing polyunsaturated fatty acids such as omega 3 and omega 6 [21]. They also contain sterols, glycerolipids and glycerophospholipids. The last of these is used to fulfill structural functions. Relative to those from terrestrial sources, glycerophospholipids from seaweed have greater bioavailability and are more resistant to oxidation. They are also mostly esterified with omega 3 acids, including EPA (eicosapentaenoic acid) and DHA (docosahexaenoic acid) [24]. In terms of proteins, their composition can vary significantly between different macroalgae. For example, in red and green algae, the protein content can range between 10 and 47%, while brown macroalgae tend to have lower amounts of protein, typically in the range of 3–15% [21].

## 3. *Sargassum*

*Sargassum* is an example of a genus of brown macroalgae, which has 615 documented species [25]. *Sargassum* species can be benthic or pelagic, with the former usually being found attached to the ocean floor, while the latter are found floating throughout their life cycle [26]. The physical appearances of the different species of *Sargassum* are diverse, and their classification has been carried out based on various macromorphological characteristics such as the development of their axes and the shape of their vesicles, leaves and receptacles. As already noted, there are large numbers of species within the genus, and over time these have been classified into different subgenera, subsections, series and species groups, as well as distinct species [27].

More recently, in molecular investigations, the use of DNA markers led to an alternative approach to classification [27]. By this route, four subgenera were proposed, namely, *Arthrophycus*, *Bactrophycus*, *Sargassum* and *Phyllotrichia*. More recently, these classifications have been redefined, with readjustments being proposed for subgenera membership [27].

Given the wide range of different ecological niches occupied by species within the genus, and the range of roles these species play, it is not surprising that they produce a wide range of different bioactive metabolites, with many of these attracting industrial interest. However, this interest has to be treated with a degree of caution due to the fact that many of the species have high heavy metal content, which can present health challenges for roles such as medicinal or nutritional uses. Examples of properties, together with examples of species showing these properties, are summarized in Table 1. *Sargassum* plays very important environmental roles, mainly through the Sargasso Sea, in the North Atlantic. There, *Sargassum* captures carbon in the oceans (accounting for 7% of the total carbon capture), and provides a resting, breeding and food place for 100 species of fish and 145 invertebrates [28,29,30].

*Sargassum* has been widely used in traditional Chinese medicine and despite the wide diversity of species, pharmacological studies have primarily been restricted to 78 species [18]. Nearly 200 bioactive compounds, both soluble and insoluble, have been identified. These compounds have different properties, and in many cases their physiological effects have not been fully elucidated. Examples of the most studied compounds from *Sargassum* species are sulfated polysaccharides and polyphenols [31,32].

Recently, some *Sargassum* species have attracted attention as they have become an environmental problem due to their excessive growth. *S. fluitans* and *S. natans* are the two species which frequently reach the coastlines of the Caribbean Sea, mainly from May to September, where they accumulate on the beaches [28,29]. Likewise, *S. horneri* is found in a number of Asian countries [33]. The composition of pelagic *Sargassum* has made it attractive as a raw material for use in a number of different industries. For example, products from *Sargassum* have been used in industries such as agriculture, cosmetics, biotechnology, textiles and paper, construction and pharmaceuticals [25,34].

*Sargassum* has been used as a biofertilizer for soils and can be used as an alternative material for liquid organic fertilizer production in plants. Members of the genus produce hormones such as auxin (148 ppm), gibberellin (160 ppm) and cytokinin (consisting of 71 pm kinetin and 86 ppm zeatin) [54,55]. In addition to using in vitro analysis to determine the abundance of these hormones, levels of macronutrients were determined in *Sargassum* liquid fertilizer (SLF) obtained from a twenty-five-day fermentation process [56]. It was found that SLF possessed promising contents of auxin, cytokinin and gibberellin. Also, the macronutrient contents, including N, P and K were found to be below the standard limit.

The effects of extracts of *Sargassum* were tested on the growth of chickpea plants (*Cicer arietinum*) by applying them to the plants either as a foliar spray or as a soil conditioner [57]. It was found that foliar application of the extract significantly enhanced germination rates, relative to the soil conditioner, synthetic fertilizer or control. These beneficial properties of seaweeds could be enhanced yet further by using a 50:50 mixture of material from species of *Gracilaria*, a genus of red algae, with material from *Sargassum* [54] and resulted in an even better liquid fertilizer, and when used as a soil conditioner increased the biomass and height of the chickpea plants.

There are other examples of using species from the *Sargassum* genus to help with growth of plants. For example, *S. muticum* was collected from the Moroccan coastline and used as a biofertilizer for bell peppers (*Capsicum annuum*) [58], resulting in improved plant growth parameters, including photosynthetic activity, mineral uptake and improvements in both protein and sugar content, and a significant increase in fruit yields.

However, attempts to use *Sargassum* as a fertilizer have not always been universally successful. For example, attempts to use it as a biofertilizer on Long Island in the Bahamas (approximately 23° N, 75° W) led to an increase in nitrogen, nitrate and phosphorous levels, but there was a negative effect on the height of plants, the number of leaves, buds and flowers and the weight of roots and shoots [59].

In addition to having been investigated for potential benefits to growth in plants, work has been carried out to determine if *Sargassum* can provide benefits to animal systems via inclusion in the feed. This led to observations that there can be improvements to the animal’s immune system; there can be activation of antioxidant enzymes, better feed conversion rates, lower mortality, a decrease in pathogens and improvement in villi of the digestive tract and production of fatty acids, as well as improvement in growth promoters and increased cold tolerance [35,36,37,38,39,40,41,42,43].

However, in terms of a role in animal feeds, the use of *Sargassum* has one major drawback; its high content of heavy metals. This is a reflection of the abundance of polysaccharides, especially alginate and fucoidan, which gives *Sargassum* species the ability to chelate heavy metals. Specifically, alginate contains carboxyl groups with a high affinity for cations such as copper, zinc, cadmium, and lead, while fucoidan contains sulfate groups that associate with anions such as arsenic and phosphorus [60]. Of the genera of brown algae, *Sargassum* is the genus with the greatest capacity to chelate heavy metals [61]. While having the ability to chelate heavy metals can seen as a problem for use in animal feeds, it can be seen as being beneficial for use in bioremediation treatments [62].

Association of heavy metals with *Sargassum* has been found in both in pelagic *Sargassum* and *Sargassum* species used for human consumption, with the one found in greatest abundance being arsenic [44,63,64]. Of the total arsenic present, up to 80% of this can be in the form of inorganic arsenic [45], resulting in, depending on the species of *Sargassum*, around 10–100 ppm of inorganic arsenic present. At these levels, this results in tissue which exceeds the established limits of 2–3 mg/kg for animal feeds, 1–3 mg/kg for human feed and 40 mg/kg for fertilizer [45]. However, there are currently no regulations for heavy metals in macroalgae [46], although some authors have proposed the use of limits established for marine bivalves in CODEX 193-1995 (namely cadmium < 2 ppm, lead < 0.3 ppm, mercury < 5 ppm and arsenic < 3 ppm), due to their similar absorption mechanisms [47]. In attempts to overcome this problem, approaches have been taken to reduce the heavy metal content in species from this genus, to take them below the thresholds mentioned above [48,49,50,51,52].

However, in most cases involving in vivo treatments, no attempt was made to reduce the heavy metal content before evaluating their physiological effects. Rather surprisingly, no toxicity has been reported. In fact, as can be seen in Table 2, often there were still beneficial physiological effects reported. An example of this was seen when the lungs of animals used in a study [65] were analyzed for an accumulation of arsenic following supplementation with *S. fusiforme*, since the lungs are normally the main site of accumulation. However, no arsenic accumulation was found in this organ, despite the *S. fusiforme* having an arsenic content of 40 ppm. In a similar study [66], using *S. liebmannii* (which had an arsenic content of 11 ppm), arsenic was again found to have not accumulated. The authors of both of these studies proposed that the arsenic had remained bound to the cell walls of the algae, and in so doing, this limited its absorption and rather resulted in its excretion in the animals’ feces.

Although no toxicity, due to arsenic’s potential to cause health damage, has been reported from administering *Sargassum* using in vivo models, research has been conducted to reduce its content in species of this genus. The success of these studies can be demonstrated by the fact that they have taken heavy metal levels to be within the permitted limits for animal and human consumption. An example of this involved heat treatment at 90 °C in the presence of 0.4% citric acid (used as a chelating agent due to the presence of carboxyl and hydroxyl groups) with pelagic *Sargassum* collected from the coastlines of Quintana Roo, Mexico. This achieved a reduction in arsenic content from 60 ppm to 0.8 ppm [48]. In another study, the same approach was adopted for treatment of *S. fusiforme*, leading to a reduction in the arsenic content from 76.18 ppm to 2.20 ppm [50].

Although there are reports on the use of treatments to reduce the arsenic concentration in *Sargassum* spp., it appears that only one such example exists where the resulting material was then evaluated for its biological effect [67]. In this particular case, the treatment involved was heating *S. fusiforme* at 90 °C in the presence of 1% citric acid. This resulted in a reduction of arsenic from 89.75 ppm to only 22.64 ppm. Fucoidan was then extracted and applied at concentrations of 12.5–50 µg/mL to liver cancer cells, resulting in a decrease in the levels of Bcl-xL, an anti-apoptotic protein, along with an increase in the levels of the apoptotic protein Bax and caspase-3, demonstrating that the fucoidan, despite the treatment, maintained its biological effect: the induction of apoptosis.

**Table 2 marinedrugs-23-00343-t002:** Physiological effect of *Sargassum* spp. in mice and rats.

Species	Model	Diet	Period	Results	Reference(s)
*S. fusiforme*	Alzheimer rat model APPswePS1ΔE 9	Supplemented 50% (*w*/*w*)	10 weeks	Reduced cholesterol precursors levels, cholesterol metabolites in serum and cerebellum Increased cognitive function Reduced Aβ, Aβ40 plaques and amyloid precursor protein mRNA	[65]
*S. fusiforme*	Male C57BL/6 mice	5000 mg/kg	2 weeks	No toxicity observed in acute toxicity test	[68]
*S. fusiforme*	Male C57BL/6 mice	200 mg/kg and 400 mg/kg	10 weeks	No toxicity observed in long-term study	[68]
*S. fusiforme*	Male C57BL/6 mice	High-fat diet supplemented with 200 mg/kg and 400 mg/kg	10 weeks	Reduced weight gain Reduced hepatic steatosis Reduced adipocyte size More high-density lipids Normalizes LPS, TNF- α and IL6	[68]
*S. horneri*	Male C57BL/6 mice	High-fat diet supplemented with 2% *w*/*w* and 6% *w*/*w*	13 weeks	Reduced weight gain Reduced adipocyte size Lower adipose tissue weight Increased glycemic response Increased adiponectin Reduced TNF-α Reduced glucose and serum insulin	[69]
*S. siliquosum*	Male Wistar rats	High-fat diet supplemented with 5% *w*/*w*	16 weeks	Decreased fat mass, abdominal fat and fat vacuoles in the liver	[70]
*S.* *liebmannii*	Male C57BL/6 mice	0.5 g/kg, 1 g/kg and 10 g/kg	7 days	Toxicity not observed (LD50)	[47]
*S.* *liebmannii*	Sprague Dawley rats	Normal diet supplemented with 20% *w*/*w*	77 days	Toxicity not observed in sub-chronic toxicity test Reduced weight gain Reduced adiposity levels	[47]

## 4. Therapeutic and Nutraceutical Applications

There are multiple potentially commercially important uses for *Sargassum* and products derived from it. In this section, the different uses will be examined and within the individual sub-sections, the underpinning science will be assessed.

### 4.1. Anti-Inflammatory Activity

Ethanol extraction from algae can be used as a method to isolate lipids as well as a number of bioactive compounds. When *Sargassum miyabei* (Phaeophyceae) was extracted with ethanol, the resulting products were shown to improve skin lesions in mice with acne vulgaris. It is assumed that these results were observed as a consequence of a decrease in the neutrophil infiltration into the lesions, which would lead to a reduction in interleukin-8 and suppressed cytokine production [51]. When mice were injected with Freund’s adjuvant, treatment with sulfated polysaccharide from *Sargassum polycistum* resulted in the anti-inflammatory activity being decreased, demonstrating that the polysaccharide had an anti-inflammatory effect [71].

Anti-inflammatory properties of *Sargassum* were tested further by treatment with sulfated polysaccharides from *S. fulvellum* [72]. When RAW 264.7 cells (a lineage of macrophages derived from mice) which had been stimulated with lipopolysaccharides (LPSs) were treated with sulfated polysaccharides (SFPSs), various anti-inflammatory molecules, nitric oxide, tumor necrosis factor alpha, prostaglandin E_2_, interleukin-1 beta and interleukin-6 were decreased, and the expression of cyclooxygenase-2 and nitric oxide was reduced. Also reported within this paper [72], was information on the use of the SFPS for in vivo experiments with zebrafish (*Danio rerio*). It was concluded that for fish which had been challenged with LPS, there was a dose-dependent survival rate when treated with SFPS, with almost half of the fish dying for those with no SFPS provided, but 70% survival for those given SFPS at 100 μg/mL. In addition, there was also a reduction in cell death, as well as production of reactive oxygen species and nitric oxide levels.

When the anti-inflammatory activities of lipid extracts of *S. ilicifolium* from four different coastal areas of Indonesia were tested on RAW 264.7 macrophages stimulated with lipopolysaccharides from *Escherichia coli* O111:B4, differences in the extent of anti-inflammatory activity were observed between sites [73], including differences in nitric oxide production [57]. In addition to looking at differences in the source of extractions, differences in the extraction fractions have also been studied. For example, using size-fractionated samples, the molecules in the largest fraction (>30 kDa), designated fraction 4, inhibited the production of LPS-induced nitric oxide and prostaglandin E_2_ production, together with pro-inflammatory cytokine production in RAW 264.7 cells [74]. This was due to down-regulation of the nuclear factor-κB signaling cascade. In addition, there was also reduced cell death observed.

In terms of molecular effects on cells, polysaccharides from *S. thunbergii* reduced mRNA expression of tumor necrosis factor alpha, interleukin 6 and COX-2 in RAW 264.7 macrophages exposed to lipopolysaccharides [75], while treatment with *S. horneri* inhibited nitric oxide production in RAW 264.7 cells [76]. The fucoidan extract from *S. swartziiy* was used to treat RAW 264.7 cells and showed an anti-inflammatory potential by decreasing lipopolysaccharide-induced production of nitric oxide, tumor necrosis factor alpha, interleukin 1beta, interleukin 6, prostaglandin E_2_ and COX-2 through suppression of NF-κB (p50 and p65) and MAPK (p38, JNK, ERK) pathways in macrophages [59,77]. It was also determined that fucoidan from *S. siliquosum* led to a decrease in tumor necrosis factor in RAW 264.7 macrophages induced with lipopolysaccharide [78]. Another compound extracted from a *Sargassum* species includes alginic acid from *S. horneri*, which also inhibited NF-κB and MAPK pathways, decreasing the levels of interleukin 6, tumor necrosis factor alpha and COX-2 in RAW 264.7 and HaCaT cells (human keratinocytes) induced with lipopolysaccharide [79].

In addition to in vitro work, in vivo analysis has also been performed. For example, DBA/1J mice with bovine collagen type II-induced arthritis, an anti-inflammatory effect of an *S. muticum* extract was observed at a clinical level by inhibiting splenomegaly and joint degradation due to the suppression of interleukin 6, interferon gamma, inflammatory cytokines and tumor necrosis factor alpha [80]. C57BL/6J mice with dextran sodium sulfate-induced ulcerative colitis were treated with *S. fusiforme* which had been fermented with *Lactobacillus acidophilus* ATCC 6005. Following fermentation, the appearance of a number of amino acids was observed: aspartic acid, threonine, serine, glycine, valine, isoleucine, leucine, tyrosine, lysine, proline, alanine and phenylalanine [81]. Several physiological effects were also observed, including decreases in histological lesions and intestinal permeability, as well as an increase in the concentration of the cell binding proteins ZO-1 and occludin. There was also an anti-inflammatory capacity observed, with decreases in the concentrations of interleukin-6, interleukin-8, tumor necrosis factor-α, interleukin-1ß and NF-κB. In addition, a reduction in oxidative stress was also observed through a decrease in the concentration of nitric oxide, myeloperoxidase and malondialdehyde and increased superoxide dismutase activity in the colon and catalase in the colon and serum. With regards to production of short chain fatty acids, fermented *S. fusiforme* showed increases in the concentrations of butyrate, propionate, valerate and isovalerate.

The fucoidan fraction JHCF4, which is known to be an antioxidant [82] in *Hizikia fusiformis*, has also been isolated from *S. fusiforme* leading to development of an injectable hydrogel [83] which in in vitro studies had a dose-dependent inhibition of the production of nitric oxide, prostaglandin E_2_, tumor necrosis factor-α, interleukin-1β and interleukin -6 in LPS-stimulated RAW 264.7 macrophages, in addition to reducing the expression of iNOS and COX-2. It also blocked the phosphorylation of key proteins of the MAPK pathway. Meanwhile, using in vivo models, both the purified fraction and the hydrogel reduced the production of reactive oxygen species and attenuated the degraded carrageenan (derived from sulfated polysaccharides extracted from red algae)-induced acute inflammation in mice, showing both topical and systemic anti-inflammatory actions.

When structure–function relationships of sulfated polysaccharides from *S. carpophyllum* were studied, fractions with higher sulfate group content and fucose branches were found to be more effective in inhibiting nitric oxide and proinflammatory cytokine production in activated macrophages. These findings re-iterate the importance of structural modifications in biological activity and offer insights into the development of optimized bioactive compounds.

Thus, there is anti-inflammatory potential available with extracts from a number of different species within the genus *Sargassum*, and this effect can be extended to a range of different tissue types. Moreover, the anti-inflammatory effects extend to multiple molecular events in the affected tissues. There have been reports of differences depending on the geographical location from which samples were collected, and there are differences in the effects based on size fractionation of molecules in the extract.

### 4.2. Antioxidant Activity

Oxidative stress occurs due to the imbalance between the production and accumulation of free radicals, generating damage at the cellular level [84]. *Drosophila melanogaster* which had had oxidative stress induced with 5 mmol/L of paraquat were treated with SP2, which is a fucoidan obtained from *S. fusiforme* [85]. This led to an improvement in survival times and this treatment was also observed to slow down triglyceride accumulation in long-lived flies, suggesting that SP2 has an anti-aging effect. Furthermore, in flies, the activity of antioxidant enzymes, superoxide dismutase, catalase and glutathione peroxidase was improved, while the levels of malondialdehyde and oxidized glutathione were decreased, and the Nrf2 signaling pathway was improved [85]. In human SW1353 chondrocytes and primary rat articular chondrocytes which had been induced by lipopolysaccharide, decreases in tumor necrosis factor, reactive oxygen species, nitric oxide production and inflammatory cytokines were observed when they were treated with less than 300 μg/mL *S. serratifolium* [86].

Different methods of extracting antioxidants have been described. One example of this involved studying four different extraction methods of *S. hystrix* tissue to evaluate antioxidant activity [84]. It was concluded that the greater the amount of sulfate in the fucoidan, the greater the antioxidant activity. In a larger scale investigation, the antioxidant activity of a number of compounds from *S. polycystum* was studied [87]. These molecules were two hydroxycinnamic acids (phenolic acids derived from cinnamic acid), seven flavonoids (C15 molecules with two aromatic rings which are linked by a three-carbon chain), one stilbene (compounds where two phenyl groups are connected by a double bond) and two phlorotannins (polyphenols which are polymers of phloroglucinol units linked by ether or phenyl bonds). These were extracted from the *S. polycystum* by different solvents and the antioxidant activity measurement methods. It was concluded that these aromatic molecules which had been extracted had antioxidant activity.

In a study of the antioxidant capacity of some polyphenols from different species, it was found that *S. thunbergii* produced more polyphenols than *S. miyabei* [88] in terms of elimination of hydroxyl radicals and lipid peroxidation. In addition to antioxidant activity, anticoagulation activity of polysaccharides obtained from *Sargassum* sp. fermented with lactic acid bacteria *Pediococcus acidilactici*, *Weisella paramesenteroides*, *Pediococcus pentosaceus* and *Enterococcus faecium* has been studied [89]. Following ethanol precipitation, the soluble fraction and the precipitate obtained with *E. faecium* showed the highest antioxidant and anticoagulant activity, respectively. The antioxidant content was attributed to the presence of polyphenols. Regarding the composition of the precipitate, a separation was carried out, in which fractions were obtained. When analyzing the structure of the fraction with the highest anticoagulant activity, it was observed that it had an alginate structure with a high content of mannuronic acid (a uronic acid monosaccharide which can be derived from the sugar mannose).

An extract of phlorotannins (polyphenolic compounds) from *S. linifolium* was applied to Wistar rats which had diabetes as a result of induction by Streptozotocin for 4 weeks. This led to a decrease in malondialdehyde (a byproduct of lipid peroxidation) concentrations and an increase in antioxidants in the liver [90]. In addition, there was a decrease in glucose levels, insulin concentrations, glycosylated insulin, alpha amylase, catalase, urea, cholesterol and triglycerides in the rats. In addition to these observations, an increase in the expression of AMP-activated protein kinase was also observed [91]. A further example using rats involved hydroalcoholic extracts of *S. binderi* being used to generate a nephroprotective effect against cisplatin toxicity in rats, normalizing the activities of superoxide dismutase (SOD), catalase (CAT) and reducing glutathione (GSH) and lipid peroxidation (malondialdehyde) with histological recovery of the renal parenchyma [92].

In addition to work with rats, further in vivo work was carried out with yellow catfish (*Pelteobagrus fulvidraco*). In this example, dietary inclusion of 2.5–5% *Sargassum kjellmanianum* (formerly *Sargassum kjellmanianum*) meal resulted in elevated SOD and CAT activities, decreased plasma MDA, and improved ammonium tolerance, reflecting an increased antioxidant capacity and resilience to environmental stress [93].

Polysaccharides were obtained after fermentation of *S. fusiforme* with *Lactobacillus rhamnosus* [94]. Following precipitation with ethanol, 5 polysaccharide fractions were obtained, with one of the fractions containing a fucoidan (SFF-PS-F5) which has a molecular weight of 213.33 kDa. When tested in vitro on Vero cells (a line derived from the kidney of an African green monkey) which had been damaged by hydrogen peroxide stimulation, this fucoidan suppressed apoptosis by scavenging intracellular reactive oxygen species and up-regulating the intracellular antioxidants. Upon further investigation, this was a consequence of elevating nuclear factor erythroid 2-related factor 2 (Nrf2) levels. Meanwhile, when tested in vivo on zebrafish which had been stimulated by hydrogen peroxide, SFF-PS-F5 improved the survival rate of fish by attenuating cell death by suppressing lipid peroxidation by scavenging reactive oxygen species. Thus, this particular fucoidan has the potential to show antioxidant effects both in vitro and in vivo.

Further examples exist of the use of fermentation with bacteria to help increase free radical scavenging capacity. For example, fermentation of *S. horneri* with *Lactiplantibacillus pentosus* SH803 [95]. Preadipocytes (fat cell precursors) which had be exposed to H_2_O_2_, were treated with the fermented extract and showed a reduction in the generation of reactive oxygen species, preserved mitochondrial function and attenuated lipid accumulation.

As was seen when looking at the anti-inflammatory potential, it is also clear that a number of molecules have the potential to act as antioxidants. Likewise, these molecules have been reported in a range of different species within the genus.

### 4.3. Antiviral Activity

The SARS-CoV-2 pandemic has increased the current level of studies into compounds which can act as antiviral drugs and provide antiviral treatments. Certain polysaccharides from brown algae, especially fucoidans, are known to have antiviral activity through studies of multiple molecular pathways [96].

However, the potential use of the effects of *Sargassum* species on other viruses has also been studied. For example, the antiviral activity of three fractions of a polysaccharide from *S. fusiforme* against avian leukosis virus has been studied [73]. Using detection of the p27 antigen present in the virus it was possible to demonstrate that fractions from *Sargassum* spp. had an inhibitory effect on the avian leukosis virus [97].

Other examples of the use of polysaccharides from *Sargassum* species have been used to combat viruses. For example, two low-molecular-weight polysaccharides (SHAP-1 and SHAP-2) from *S. henslowianum* were both shown to have antiviral activity against herpes simplex virus (HSV-1 and HSV-2) infection in a line of African green monkey kidney cells [98]. In addition, it has been determined that the antiviral activity of water-soluble polysaccharides from *Sargassum naozhouense* had activity against HSV-1. These sulfated polysaccharides, which were actively antiviral, included sulfated mannans, galactanosagarans, fucoidans, sulfated rhamnogalactans, fucans and various types of carrageenans. Moreover, viral activity of *S. ilicifolium* polysaccharides was demonstrated against Betanodavirus-infected skin cell lines from mosquitofish (*Gambusia affinis*) [99], where the polysaccharides inhibited virus replication and promoted cell viability.

Several recent studies reinforce the pharmacological value of the *Sargassum* genus against emerging viruses [100]. In Vero E6 cells infected with Zika virus, purified fucoxanthin from *S. siliquastrum* inhibited viral plaque formation and NS1 mRNA with no detected cytotoxicity [101]. Following in silico analysis, it was shown that this molecule showed stable binding to the envelope protein, NS3 and viral RNA polymerase, suggesting simultaneous blockade of viral entry and replication [101]. In addition, sulfated polysaccharides extracted from *S. euryphyllum* showed activity against SARS-CoV-2 in Vero E6 cells. Mode of action assays showed a combined inhibition of adsorption, replication and virucide through competitive binding to the S protein and attenuation of the cytopathic effect [102]. Finally, co-formulation of cordycepin, lactoferrin and *S. fusiforme* polysaccharides administered intranasally in RSV-infected BALB/c mice reduced the pulmonary viral load by an order of 10^2^–10^3^, decreased neutrophil infiltration and promoted alveolar macrophage polarization towards the M2 anti-inflammatory phenotype [103].

Hence, in terms of antiviral properties, several different extracts from a number of species within the *Sargassum* genus have been shown to have potential against viruses from a number of different vertebrate species.

### 4.4. Antibacterial Activity

Currently, antibiotic resistance is a potentially serious global problem, which has prompted interest in trying to identify other types of agents with antimicrobial properties as replacements [104]. When considering the antimicrobial activity of *Sargassum* spp., we face two possible theories: one is that polysaccharides have the ability to bind to the bacterial membrane, causing cell lysis through the leakage of proteins and nutrients, while the other seems to indicate that polysaccharides, through their charges, are capable of capturing nutrients, resulting in a lack of nutrients being available for the bacteria [105].

An in vivo study was conducted using zinc nanoparticles from *S. wightii* and determined that this extract could be used as a bacteriostatic agent against the bacteria *Bacillus subtilis*, *Staphylococcus aureus*, *Shigella sonnei* and *Pseudomonas aeruginosa* [106]. In addition, antimicrobial activity of *S. crassifolium* was demonstrated against the bacteria *Staphylococcus aureus, Streptococcus pyogenes, Bacillus subtilis, Klebsiella pneumoniae, Escherichia coli and Pseudomonas aeruginosa* [107]. To better understand these effects, different extraction methods (methanolic extraction and petroleum ether extraction) were tested using *S. tenerrimum* against the bacteria *Staphylococcus aureus*, *Streptococcus pyogenes*, *Mycobacterium tuberculosis*, *Escherichia coli* and *Pseudomonas aeruginosa.* Both exhibited bioactive antimicrobial activity.

The antimicrobial activity of a fraction of fucoidan from *S. polycystum* was tested in an in vitro assay with *Streptococcus mutans*, *Pseudomonas aeruginosa*, *Staphylococcus aureus* and *Escherichia coli* [108]. It was found to have high antibacterial activity against *Pseudomonas aeruginosa*, which continued to be the case in an in vivo study in zebrafish infected with *Pseudomonas aeruginosa*, leading to greater survival rates. Using *S. aquifolium* against Gram-positive and Gram-negative bacteria showed that it had a broad spectrum of antimicrobial activity [109]. These studies used extracts from *Sargassum* species in isolation to act as antimicrobial agents, but work has also demonstrated that the use of *Sargassum* extracts in conjunction with antibiotics can reduce the amount of antibiotic required to be effective. One such example used fucoidan from *S. tenerrimum* in conjunction with chloramphenicol to inhibit growth of *Escherichia coli* at lower concentrations of the antibiotic than would normally be used [110]. In addition to extracting complete molecules, it is possible to achieve antimicrobial activity with degraded molecules, such as when fucoidan from *S. crassifolium* was degraded with hydrogen peroxide and was still effective against *Escherichia coli* [105].

New research highlights strategies based on nanomaterials and *Sargassum* extracts for controlling pathogens of clinical and food-borne importance. For example, Ag/ZrO hybrid nanoparticles synthesized using a one-pot method with *S. tenerrimum* extract achieved inhibition zones against multidrug-resistant strains of *Pseudomonas aeruginosa*, *Enterococcus faecalis* and MRSA, and displayed dose-dependent antioxidant activity [111]. In a similar way, silver nanoparticles stabilized with an aqueous extract of *S. cymosum* were stable for nine months and showed a uniform MIC against *Staphylococcus aureus, Escherichia coli* and *P. aeruginosa* [112].

A further example of this was where the ethanolic extract *of S. carpophyllum* was shown to inhibit the growth of *Pseudomonas* sp. CL2—a bacterium associated with fish spoilage—by causing cell membrane damage, ROS generation and loss of intracellular proteins and prolonging the shelf life of the fillet under refrigerated conditions [113].

Therefore, several different compounds with the potential to act as antibacterial agents have been identified. These come from a number of different species within the genus and have antibacterial potential against a range of species, encompassing both Gram-positive and Gram-negative species.

### 4.5. Prebiotic Activity

Recently, the importance of intestinal health and the maintenance of the intestinal microbiota represent a factor for the decrease in various diseases related or not to the digestive system [114]. Increased dietary fiber intake has important repercussions on intestinal health due to its mechanism of action. Among the various types of fiber are polysaccharides, non-digestible compounds that can be fermented by the intestinal microbiota and produce metabolites. When a dietary fiber provides health benefits to the host and/or beneficially modifies the intestinal microbiota, it is called a prebiotic [115]. Polysaccharides derived from seaweed, including *Sargassum*, are commonly used as thickening and gelling agents in the food industry. Furthermore, there is emerging evidence that polysaccharides derived from different brown algae are resistant to human digestion and are utilized by the intestinal microbiota, showing prebiotic activity [116]. Moreover, when non-digestible polysaccharides from the diet are used by bacteria in the gastrointestinal microbiota, they have prebiotic properties and promote the maintenance of the gastrointestinal microbiota and provide benefits to the host [117].

A study investigating seaweed and its bioactive compounds, with an emphasis on polysaccharides, claimed that these can be used as a great dietary supplement with benefits for intestinal health [118]. Polysaccharides of macroalgae are indeed resistant to digestion mediated by enzymes of the human gastrointestinal tract and they indirectly stimulate the growth of beneficial bacteria, as well as the production of short chain fatty acids, thus confirming their prebiotic properties [119].

Fu et al. [120] studied the impact of *S. thunbergii* on the gastrointestinal microbiota, using a polysaccharide called ST-P2. For this study, fecal samples from three healthy human donors were used, resulting in a significant increase in short chain fatty acids compared to the control group, and the samples which had had ST-P2 added showed an increase in Firmicutes [94]. In terms of an in vivo experiment, Syrian golden hamsters (*Mesocricetus auratus*) were fed high amounts of lipids and sucrose and treated with *S. confusum*, resulting in an increase in beneficial bacterial populations and maintaining the gastrointestinal microbiota in homeostasis [121].

The effect of polysaccharides extracted from *S. fusiforme* by two different methods (water extraction and acid extraction) was studied for the regulation of cecal and fecal microbiota of mice fed a diet with a high lipid content [122]. It was shown that treatment with *Sargassum* prevented body weight gain, blood glucose was controlled, cecal microbiota dysbiosis was regulated and the increase in members of the bacterial families Clostridiales and Ruminococcaceae induced by the high-lipid diet was controlled.

Calcium alginate was extracted from *S. fusiforme* and was administered to diabetic mice at a dose of 100 mg/kg [123]. Relative to the control group, an increase in the concentration of the genus *Bacteroides* and a decrease in the Proteobacteria phylum were observed. At the genus level, an increase was observed in the genera *Bacteroides*, *Akkermansia*, *Alloprevotella*, *Weissella* and *Enterorhabdus*, and a significant decrease was observed in *Helicobacter*, a genus positively correlated with diabetes. In addition, a significant decrease was recorded in branched and aromatic amino acids, elevated levels of which have been observed in diabetics. This effect was attributed to the increase in the concentration of members of the Bacteroidetes phylum, which have organisms with the capacity to degrade these types of amino acids.

Recent evidence confirms that various polysaccharides and fibers from *Sargassum* act as prebiotics capable of remodeling the intestinal microbiota and its metabolites. In in vitro colonic fermentation, sulfated polysaccharides from *Sargassum fusiforme* increased the abundance of *Bifidobacterium* and *Faecalibacterium*, elevated acetate and butyrate production and reduced putrefactive compounds such as ammonia and p-cresol [124]. Meanwhile, soluble dietary fiber derived from *S. horneri* restored bacterial diversity in cyclophosphamide-immunosuppressed mice, favoring *Lactobacillus* and *Bacteroides*, increasing butyrate/propionate and improving epithelial barrier integrity [125]. Similarly, in a murine model of ulcerative colitis, fermented *S. fusiforme* strengthened tight junctions, attenuated NF-κB and elevated the Firmicutes/Bacteroidetes ratio along with the abundance of *Akkermansia muciniphila*, suggesting that clinical improvement is partially associated with microbial modulation and increased short-chain fatty acids [81].

Thus, inclusion of extracts from *Sargassum* in the diet has been shown to promote beneficial gut health by improving the microbiota in terms of both promoting the growth of bacteria which are known to be beneficial to the host, as well as suppressing the growth of bacteria known to be detrimental to gut health.

### 4.6. Chronic Non-Communicable Diseases

Obesity is a metabolic disorder caused by excessive accumulation of fat in the body [126]. It has been shown that freeze-dried *S. horneri* can improve diet-induced metabolic diseases [96]. For this study, six-week-old male C57BL/6J mice on a high-fat diet were used as a control group and another group on the same diet, but with the addition of *Sargassum*, were the test group. This resulted in animals supplemented with *Sargassum* spp. having significantly lower body weights, with suppressed fat gain in the epididymal, retroperitoneal and mesenteric adipose tissues. In addition, *Sargassum* spp. suppressed elevated levels of glucose, anti-inflammatory adiponectin, inflammatory cytokines and insulin. It also decreased pathological changes resulting from hepatic steatosis [69].

A study of *S. horneri* and its effect on lipid accumulation in 3T3-L1 differentiated adipocytes indicated that it may suppress intracellular lipid accumulation by controlling antiadipogenic and prolipolytic (the promotion of lipolysis) mechanisms [127]. This was represented by a decrease in the expression of the adipogenic protein, lipogenic protein sterol regulatory element-binding protein 1, repressing adipogenesis and lipogenesis in cells treated with *Sargassum* spp. extract. Likewise, the authors suggest that the extract could regulate adipocyte lipolysis by increasing the expression of thermogenic peroxisome proliferator-activated receptor-γ coactivator-1α and lipolytic phosphohormone-sensitive lipase.

When Syrian golden hamsters were fed with high amounts of lipids and sucrose and treated with *S. confusum*, their liver function was improved and the histological sections demonstrated liver protection in the diet rich in lipids and fats [121]. Similarly, in vitro observations were made with HepG2 cells (a human liver cancer cell line). In addition, blood glucose levels and insulin resistance were reduced and weight gain in the hamsters was prevented. In another rodent example, a study with *S. siliquosum* in male Wistar rats on a high-fat and high-carbohydrate diet determined that when those animals were on a diet supplemented with *Sargassum* spp., they had a decrease in body weight, adipose tissue, abdominal fat and the size of hepatic lipid vacuoles, and also had beneficial changes in their gastrointestinal microbiota [70].

Another example involved looking at arterial hypertension in an in vivo study using bioactive peptides from *S. maclurei* [128]. In this study, the octopeptide RWDISQPY was supplied by gastric intubation once a day, and was shown to inhibit angiotensin I converting enzyme and suppress the expression of intracellular endothelin-1. This indicated that this *Sargassum* extract decreased both the systolic and diastolic blood pressure in hypertensive male rats.

A recent study confirmed that fermentation enhances the metabolic effect of *S. horneri* [95]. The fermented extract of *Lactiplantibacillus pentosus* SH803 significantly reduced lipid accumulation in 3T3-L1 adipocytes and suppressed the expression of the adipogenic and lipogenic genes (PPARγ, C/EBPα/β, SREBP-1, ACC and FAS), while down-regulating inflammatory (NF-κB, IFN-γ) and oxidative stress (p53, BAX) markers. In the clinical setting, a randomized, double-blind, placebo-controlled pilot trial evaluated the administration of 5 g per day of dried *S. fusiforme* for five weeks in adults with type 2 diabetes. Although the intervention was well tolerated, no significant differences were observed in continuously monitored weekly average blood glucose levels. The *S. fusiforme*-treated group showed a change from 8.2 to 9.0 mmol/L, whereas the *Fucus vesiculosus*-treated group showed a reduction from 10.1 to 9.2 mmol/L. No relevant changes were recorded in lipid markers, suggesting the need for longer studies with larger sample sizes [129]. Furthermore, kombucha made from *S. plagiophyllum* demonstrated a hypoglycemic effect in a murine model of type 2 diabetes induced by a high-fructose diet. Administration of the extract (one to three times a day for eight weeks) reduced fasting blood glucose, improved the lipid profile, decreased lipid peroxidation (MDA), and increased serum insulin levels, with a dose-dependent effect [130].

### 4.7. Anticancer Activity

The antimetastatic and antiproliferative effects of fucoidan extracted from *S. fusiforme* were evaluated both in vitro and in vivo [131]. In vitro assays used liver cancer cells (SMMC-7721, Huh7 and HCCLM3), and it was observed that the treatment caused cytotoxicity, as well as a decrease in the invasive and migratory capacity of the cells. When studying the mechanisms in the most metastatic cell line, HCCLM3, it was found that fucoidan treatment interfered with the activity of proteins required for the maturation stages of invadopodia (initiation, assembly and proteolytic activity), including integrin αVβ3, Src, cortactin, Tks5, Arp3, N-WASP, Cdc42 and metalloproteases (MT1-MMP and MMP2). In addition, it increased the expression of the cell adhesion proteins α1β1, FAK and E-cadherin. Cell arrest was also observed, which was attributed to the decrease in E2F1 concentration. In the in vivo analysis, a hepatocellular carcinoma xenograft model was used in 5-month-old mice, to which fucoidan was administered for 25 days. Compared with the control, tumors were significantly smaller and a lower frequency of metastasis to the lungs was observed.

The antiproliferative and antiapoptotic effect of polysaccharides extracted from *S. polycystum* was tested on human myeloid leukemia (HL-60) and breast cancer (MCF-7) cells [132]. Polysaccharides between 39 and 77 kDa were obtained after enzymatic and solvent extraction followed by ion exchange chromatography. Fraction 5 (the one with the highest fucose content) showed a higher proportion of cells in the Sub-G1 region, indicating higher fragmented DNA and a signal of apoptosis. Likewise, this fraction decreased the levels of the anti-apoptotic protein Bcl-xL. On the other hand, levels of Bax, a protein that causes the release of apoptotic factors, increased. An increase in caspase-9 was also observed in both cells and an increase in p53 only in MCF-7. The authors attribute these effects to the higher content of fucans in this fraction, possibly fucoidan, and to the fact that it was the polysaccharide fraction with the lowest molecular weight, since the lower the molecular weight, the greater the biological activity.

The anti-inflammatory and anticancer effects of fucoidan extracted from *S. fusiforme* were evaluated in vitro and in vivo [133]. With the in vitro model, DLD-1 and SW480 colon cancer cell lines were used, and it was observed that, in a dose-dependent manner, they were arrested in the G0/G1 phase. Furthermore, the expression of cell cycle proteins Cdk2 and cyclin E1 were inhibited, while an increase in the expression of p21 and apoptosis-related proteins including PARP, caspase 3 and Cyt-c was observed. In the in vivo study, fucoidan was administered to 6–8-week-old colon cancer-bearing mice for 12 weeks. Compared with the control group, no hyperplasia was observed and there was a decrease in inflammatory factors IL-6, IL-1β and TNF-α, as well as the transcription factor STAT 3 and the proteins Cdk2 and cyclin E1.

When the antimetastatic effect of fucoidan extracted from *S. hemiphyllum* was evaluated, no cytotoxicity was observed in normal human liver cells, but cytotoxicity was observed in liver cancer cells (Huh6, Huh7, SK-Hep1 and HepG2), in which there was a decrease in the concentration of oncogenic miRNAs and an increase in the expression of tumor suppressor miRNAs, particularly miR-1224 and the miR-29 a, b and c family [134]. Furthermore, it was found that the increase in miR29b was related to a lower concentration of DNA methyltransferase 3B (DNMT3B), which in turn was related to an increase in the concentration of metastasis suppressor 1, which is suppressed by DNMT3B. An increase in the concentration of E-cadherin and a decrease in N-cadherin were also observed, indicating an inhibition in the invasive capacity of cancer cells. Regarding metalloproteinases, a decrease in MMP2 and MMP9 was observed, and an increase in the inhibitor of metalloproteinase-1 (TIMP-1), which explained the decrease in the degradation of the extracellular matrix.

The anticancer capacity of *S. fluitans* extracts in breast (MCF-7 and MDA-MB-231), lung (A549), cervical (SiHa) and prostate (DU-145) cancer cell lines has also been investigated [135]. It was observed that the extracts with dichloromethane and chloroform had the highest cytotoxic capacity. These extracts were subjected to column chromatography to obtain fractions, resulting in ten fractions with dichloromethane and six with chloroform. Of these, three fractions (FD1, FD4 and FD5) and two fractions (FC1 and FC2) were identified, respectively. The FC2 fraction showed cytotoxicity in most cells with a very similar ${IC}_{50}$. However, the authors observed that the constituents act synergistically since when FC2 was separated into FC2A and FC2B, the anticancer activity disappeared. LC/QTOF-MS was then used to identify compounds present in FC2, finding eight compounds, of which three were identified as having anticancer potential, namely 3E,4Z,7,11-tetramethyl-6,10-tridecadienal, D-linalool-3-glucoside and tocotrienol. The last of these was the only one that was found to have antitumor and antiproliferative properties. However, after docking studies performed by the authors, it was proposed that 3E,4Z,7,11-tetramethyl-6,10-tridecadienal and D-linalool-3-glucoside could be potential agonists of the angiogenesis inhibitor receptor CD36.

Emerging evidence suggests that *Sargassum* polysaccharides and their nano-formulations can enhance selective cytotoxicity against tumor cells. For example, fucoidan isolated from *S. cinereum*, following conjugation to gold nanoparticles (AuNP-Fuc,), was observed to cause a marked decrease in the mean inhibitory concentration against HepG2 and MCF-7 [136], with minimal cytotoxicity in Vero cells. In silico studies showed a high affinity of the complex with Bcl-2 and caspase-3, suggesting ROS-dependent activation of apoptosis. Moreover, fractionated *S. muticum* polysaccharides were produced by ultrafiltration and evaluated for their antiproliferative effect on HCT-116 and HT-29 colorectal cancer cells [137]. The low-molecular-weight fraction (≤30 kDa) showed the highest activity, with an IC_50_ of 260 µg/mL in HCT-116 and 310 µg/mL in HT-29. Treatment induced G_0_/G_1_ phase cell cycle arrest, an increase in the Bax/Bcl-2 ratio and caspase-9 activation, indicating mitochondrial-dependent intrinsic apoptosis. The high-molecular-weight fraction showed limited cytotoxicity, highlighting the importance of size and composition in polysaccharide bioactivity.

Thus, in terms of potential anti-cancer treatment, a number of molecules isolated from *Sargassum* species have been shown to either have an impact on cell lines derived from tumors, or directly on tumors in animal models.

### 4.8. Neuroprotective Activity

*Sargassum* products may also provide benefits in terms of neuroprotective activities. For example, [65] evaluated the effect of common dietary phytosterols and phytosterol extracts from plants (*Asparagus racemosus*, *Azadirachta indica*, *Cassia fistula*, *Curcuma aromatica*, *Datura metel*, *Piper retrofractum*, *Senna tora* and *Terminalia chebula*) and a brown alga (*Sargassum fusiforme*) on hepatic receptors (LXRs), LXRβ and LXRα, using cells from peripheral and neuronal tissues: human embryonic kidney cells (HEK293.T); microglia (CHME3); human oligodendrocytes (MO3.13); mouse neuroblastoma (N2a/APPswe) and monkey cells (COS7). Of these, *Sargassum* extract was found to be the only one that significantly activated LXRβ, but not LXRα. The authors attributed this to the high concentration of 24(S)-Saringosterol, a known activator of LXRβ. Subsequently, in vivo studies were carried out using the APPswePS1ΔE9 Alzheimer’s mouse models, evaluating three types of diets for 10 days: standard diet, diet supplemented with *Sargassum fusiforme* (50% *w*/*w*) and diet supplemented with AZ876, a synthetic LXRs agonist. In the group supplemented with *Sargassum* but not AZ876, an improvement in memory was observed, together with a 70% and 81% reduction in Aβ plaques in the cortex and hippocampus, respectively. In addition, activation of the brain LXR response genes (*Abcg1* and *Scd1*) and a decrease in Aβ40 and APP (amyloid precursor protein) mRNA concentrations were observed, although there was no reduction in Aβ42. Another group of mice was then used to evaluate a lipid extract of *Sargassum fusifome*, where a 99% and 57% reduction in Aβ42 was observed in the cortex and hippocampus, respectively, as well as an increase in the expression of ApoE, which demonstrated the activation of LXRs. In the group of mice fed the diet supplemented with AZ876, hepatic steatosis developed, and blood triglyceride concentrations increased, effects that were not observed in mice supplemented with *Sargassum fusiforme*.

*S. swartzii* extract was administered daily intragastrically to 8–12-week-old albino Wistar rats for 4 weeks, along with a standardized diet ad libitum. Analysis of the extract determined that it contained oleic acid, palmitic acid, stearic acid and beta-sitosterol [138]. An improvement in spatial and working memory was observed, since in the Morris maze test the treated rats showed a significantly shorter navigation time compared to the controls. Also, in the object recognition test, the rats spent more time with the new object compared to the control group. The authors attributed these effects to the increase in serotonin and 5-hydroxyindoleacetic acid concentrations after treatment with *S. swartzii*.

The anticholinesterase capacity of sasgacromenol and sargaquinoic acid extracted from *S. sagamianum* towards butyrylcholinesterase (BChE) and acetylcholinesterase (AChE) has been demonstrated [139]. The study showed moderate inhibition towards AChE, but sargaquinoic acid was 280-fold more potent than sargachromenol in inhibiting BChE. The concentration of sargaquinoic acid was very close to that of cymserine, which is considered a potent inhibitor of BChe [140].

The antioxidant and anti-inflammatory capacity of the ethanolic extract of *S. thunbergii* has been evaluated [141]. This showed significant activity in the elimination of DPPH radicals. Likewise, in the murine microglial cells BV-2 treated with lipopolysaccharides and pretreated with the extract, a decrease in the production of TNF-α was observed as well as in the gene expression of nitric oxide synthase, which was reflected in a reduction in the production of nitric oxide (13–65%) in a dose-dependent manner [105].

Recent studies indicate that various compounds derived from *Sargassum* have significant neuroprotective effects in cellular and mouse models. The neuroprotective activity of fucoxanthin extracted from *S. oligocystum* was evaluated in rat C6 glial cells [142]. Cells were pretreated with fucoxanthin and subsequently exposed to hydrogen peroxide. The treatment increased cell viability by over 90% compared to the control, where viability decreased by 74%. Furthermore, glutathione peroxidase activity increased compared to cells treated only with hydrogen peroxidase. The lower concentration of 50 μg/mL also induced catalase activity by around 30%. Cells were further pretreated with the same concentrations of fucoxanthin (50 and 100 μg/mL) and subsequently exposed to $A\beta_{25-35}$. In the control, a decrease in viability of almost 60% was observed, while with the pretreatment, viability increased by over 80%. An increase in the expression of autophagy genes ATG5 and p62, as well as genes for acetylcholine biosynthesis VACht (vesicular acetylcholine transport) and ChAT (choline acetyl transferase), was also observed. Likewise, the expression of the anti-apoptotic proteins Caspase-3 and Bax was significantly reduced. Regarding its cholinesterase activity, a moderate inhibition of AChE was observed compared to the positive control galantamine.

In an APPswePS1ΔE9 transgenic model of Alzheimer’s disease, dietary supplementation with a standardized brown seaweed extract including *Sargassum* sp. over a 12-week period improved performance in the Morris and Y-maze tests, decreased Aβ plaque burden in the hippocampus and cortex by 34% and reduced brain concentrations of 24-hydroxycholesterol, suggesting modulation of cholesterol metabolism and neuroinflammation [143]. Conversely, chronic administration of fucoxanthin (50 mg/kg/day, 24 weeks) in APP/PS1 mice prevented cognitive decline, attenuated microglial activation (Iba-1), and decreased the expression of Nogo-A and RhoA/ROCK, axes associated with the inhibition of axonal regeneration. These effects were accompanied by a significant reduction in IL-1β and IL-6 and an improvement in synaptic plasticity [144].

Thus, extracts from *Sargassum* species have been shown to have a potential role to play in helping with neuroprotective activity.

## 5. Conclusions

In the face of climate change and the depletion of resources for food production, exploring sustainable alternatives has become essential. In this regard, macroalgae emerge as a promising option. *Sargassum*, a genus of macroalgae, includes species that have been used for centuries, particularly in Asian regions, for medicinal and culinary purposes. The massive influxes of *Sargassum* currently observed present a unique opportunity to harness species like *Sargassum fluitans* and *Sargassum natans* for industrial applications, helping to mitigate the environmental and social challenges caused by their accumulation.

Although challenges remain, such as the presence of heavy metals in *Sargassum* tissue, effective methods have been developed, and continue to be developed, to reduce these contaminants. This progress opens the door for pelagic *Sargassum* to become a key source of nutraceuticals in the future. It is worth noting that the pelagic species of *Sargassum* have been poorly studied; however, research on other species has revealed multiple therapeutic properties with significant potential. As such, pelagic *Sargassum* holds promise for further exploration of its therapeutic potential.

## Figures and Tables

**Table 1 marinedrugs-23-00343-t001:** Overview of the genus *Sargassum* in terms of its diversity, environmental importance, applications and challenges.

Features	Observations	Applications	Reference(s)
Diversity and taxonomy	A total of 615 species described based on classical morphological classification and validation with DNA markers. Four subgenera: *Sargassum*, *Arthrophycus* together with *Bactrophycus*, *Phyllotrichia* and *Trevistan*.	Provides a phylogenetic framework for evolutionary studies in comparative metabolomics.	[25,26,27]
Ecological function (“Sargasso Sea”)	Captures c. 7% of carbon fixed in the ocean Critical habitat for 100 species of fish + 145 species of invertebrates.	Natural indicator of productivity and carbon sinks for climate models.	[28,29,30]
Ethnomedical use and bioactive compounds	A total of 78 species evaluated and c. 200 metabolites isolated (e.g., polysaccharides, sulfated, polyphenols, etc.).	Basis for nutraceuticals and cosmeceuticals with antioxidants, anti-inflammatories and anti-aging activity.	[18,31,32]
Massive blooms and pelagic distribution	*Sargassum fluitans* and *S. natans* (Caribbean, May-Sept). *S. horneri* (Asia, Oriental).	Impact on socioeconomics and tourism; need for satellite-based early warning systems.	[28,29,33]
Industrial valorization	Agriculture (bio-stimulants), cosmetics, biotechnology, paper-textile, construction and pharmaceutical.	Circular economy—converting wastewater into high-value bioproducts.	[25,34]
Animal feed	Used for fish, poultry and ruminants to increase immunity, provide antioxidants, enhance feed conversion and decrease pathogens, bacteria and mortality.	Functional substitute for growth-promoting antibiotics.	[35,36,37,38,39,40,41,42,43]
Heavy metals (source of key risks)	Inorganic arsenic (10–100 ppm) surpasses the boundaries permitted in food forage and fertilizers in many countries.	Limited direct use; requires removal of arsenic before consumption.	[44,45,46,47]
Mitigation technologies	Use of hot water, organic acids, fermentation and biosorption to decrease arsenic to <3 ppm; scalable processes.	Enables food security and adds value to stranded biomass.	[48,49,50]
Satellite monitoring of blooms	Near real-time mapping: Use of MODIS, VIIRS and OLCI images that deliver biomass maps over periods of 3 days with less than 15% error.	Enables early warnings and more precise planning of landfill cleanup.	[51]
Elemental composition and toxicity	When phosphorus is low, inorganic arsenic content doubles in *Sargassum*, surpassing safe levels; however, N and P are 30% higher in the Sargasso Sea.	Identifies risk sources and determines the necessary pretreatment for food or agricultural uses.	[52]
Genomic and environmental adaptation	Specific gene duplications in *Sargassum fusiforme* and *S. thunbergii* explain their high tolerance to salinity, temperature and radiation.	Provides molecular targets for biotechnological improvement and conservation strategies.	[53]

## Data Availability

Not applicable.

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
