# Peer review of "Therapeutic and Nutraceutical Potential of Sargassum Species: A Narrative Review"

_marinedrugs, 2025, doi:10.3390/md23090343_

Round 1
Reviewer 1 Report
Comments and Suggestions for Authors
The manuscript entitled “Therapeutic and Nutraceutical Potential of Sargassum Species: A Narrative Review” provides an account pertaining to the potential biological properties of extracts and types of substances obtained from macroalgae of the genus Sargassum.
The account is very general in nature. The authors mention several compounds and compound classes but no chemical structures or Structure-Activity Relationships (SARs) are provided in this review.
As such, I find this manuscript of very limited interest to the vast majority of organic, medicinal and natural product chemists. Hence, I can only recommend that it is promptly rejected.
Author Response
I appreciate your comments and time in reviewing the article.

Reviewer 2 Report
Comments and Suggestions for Authors
This narrative review discusses the remarkable physiological effects of various compounds derived from Sargassum species, highlighting their positive role in addressing global challenges. While the manuscript shows significant merit, several areas require improvement prior to publication.My review comments are as follows:
Keywords
Line 29: The genus Sargassum needs to be italicized.
Algae
Line 55: Not all seaweed species are 60 meters long.
Sargassum
This section requires a supplementary introduction on the application of Sargassum resources as agricultural fertilizer.

Author Response
This narrative review discusses the remarkable physiological effects of various compounds derived from Sargassum species, highlighting their positive role in addressing global challenges. While the manuscript shows significant merit, several areas require improvement prior to publication.My review comments are as follows:
Keywords
Line 29: The genus Sargassum needs to be italicized.
changes were made
Algae
Line 55: Not all seaweed species are 60 meters long.
Yes, it is true, as indicated, it can reach up to 60 meters.
Sargassum
This section requires a supplementary introduction on the application of Sargassum resources as agricultural fertilizer.
We will work on supplementation
Supplementary information has been added lines 146-175.

Reviewer 3 Report
Comments and Suggestions for Authors
Major modifications are necessary for this work to be considered suitable for publication. This manuscript summarized the therapeutic and nutraceutical potential importance of seaweed and algae, and analyzed the bioactivities of compounds isolated from members of genus Sargassum. However, this review lacks novelty; the existing literature already includes some similar articles, and this manuscript does not provide any new insights to the field. Additionally, the manuscript does not provide a comprehensive, critical, and deep discussion of this topic. In addition, the references throughout the manuscript require updating, as nearly all cited works are from five or more years ago. For the manuscript to be considered for publication, the authors must provide a more in-depth discussion and analysis, citing relevant literature from the past five years.
(1) The article provides a comprehensive review of the various bioactivities of Sargassum species; however, it lacks an in-depth discussion of the latest research advancements (such as those from 2023-2025). It is recommended to supplement the article with recent research data.
(2) Line 442, Although the article summarizes the potential applications of Sargassum, it lacks depth in the mechanistic studies. For instance, the description of the molecular pathways related to its anticancer activity is rather brief. It is suggested to include a detailed analysis of key signaling pathways (such as NF-κB and MAPK) and their specific mechanisms of action.
(3) The conclusion section is superficial and does not adequately summarized the practical applications of the research findings. It is recommended to expand the conclusion to outline the specific application directions of Sargassum in pharmaceuticals and the food industry, as well as to highlight future research priorities.
(4) The article lacks supporting figures, such as chemical structures of bioactive compounds or schematic diagrams of their mechanisms of action. It is advisable to add 1-2 summary figures or tables that visually represent the structures of key compounds or their active mechanisms.
(5) Line 140, While the article mentions the heavy metal contamination issues associated with Sargassum, it does not systematically analyze the advantages and disadvantages of different detoxification technologies. It is recommended to include a section discussing the efficiency comparisons of physical, chemical, and biological methods for heavy metal removal, and to cite the latest technologies.
Author Response
I appreciate your comments and we will work on supplementation
(1) The article provides a comprehensive review of the various bioactivities of Sargassum species; however, it lacks an in-depth discussion of the latest research advancements (such as those from 2023-2025). It is recommended to supplement the article with recent research data.
Recent advancements have been added in different sections as follows:
Fertilizer information: paragraph in lines 146-175.
Anti-inflammatory activity: paragraph in lines 294-307.
Antioxidant activity: paragraph in lines 367-377.
Antiviral activity: paragraph in lines 404-418.
Antibacterial activity: 457-469.
Prebiotic activity 519-530.
Chronic non-communicable diseases: 573-590.
Anticancer, 657-671
Neuroprotective activity; 722-751.
(2) Line 442, Although the article summarizes the potential applications of Sargassum, it lacks depth in the mechanistic studies. For instance, the description of the molecular pathways related to its anticancer activity is rather brief. It is suggested to include a detailed analysis of key signaling pathways (such as NF-κB and MAPK) and their specific mechanisms of action.
The work has been described in greater depth.
(3) The conclusion section is superficial and does not adequately summarized the practical applications of the research findings. It is recommended to expand the conclusion to outline the specific application directions of Sargassum in pharmaceuticals and the food industry, as well as to highlight future research priorities.
The work has been expanded.
(4)The article lacks supporting figures, such as chemical structures of bioactive compounds or schematic diagrams of their mechanisms of action. It is advisable to add 1-2 summary figures or tables that visually represent the structures of key compounds or their active mechanisms.
Two tables have been added. Table 1, lines 123-125; Table 2, lines 230-231.
(5) Line 140, While the article mentions the heavy metal contamination issues associated with Sargassum, it does not systematically analyze the advantages and disadvantages of different detoxification technologies. It is recommended to include a section discussing the efficiency comparisons of physical, chemical, and biological methods for heavy metal removal, and to cite the latest technologies.
Recommendation was done as follows:
Lines 180-184, this paragraph was added: ¨because the presence of polysaccharides, especially alginate and fucoidan, gives Sargassum spp. the ability to chelate heavy metals. Alginate contains carboxyl groups with a high affinity for cations such as copper, zinc, cadmium, and lead, while fucoidan contains sulfate groups that associate with anions such as arsenic and phosphorus.¨
Also, supplementary information has been added in lines 198-227, and table 2 has been added.
Finally, references and abbreviations were checked and added. Also, grammar and spelling has been checked, minor corrections have been done as follows:
Line 33, word ´mainly´ was deleted and ´places such as´ was added.
Lines 49, 50 ´Activity´ was changed for ´activity´.
Line 60, a (,) was changed for ´are´.
Line 73, word ¨colonoic¨ was deleted and ¨short chain fatty acids¨ was added.
Line 87-88, e.g., was added.
Line 117, word ´discovered ‘was replaced for ´proposed´.
Line 129, words ´accounting for´ were added.
Line 176, words ´tried to be implemented were deleted and ´tested’ was added.
Line 247, word ´in´ was replaced for ´as´.
Line 271, word ´an´ was added
Line 276, word ´studied’ was replaced for ´observed´
Line 346, word allowed was deleted.
Line 347, -ing was changed for -ed.
Line 394, -1 was changed to superscript.
Line 297, preposition ´to´ was replaced for ´to´.
Line 447, word activity was added.
Line 515, chain was deleted.

Round 2
Reviewer 3 Report
Comments and Suggestions for Authors
ok
Author Response
The new revision has been carried out based on the observations made by the academic editor. I hope the changes made are those required for publication.